# Homotypic and heterotypic immune responses to Omicron variant in immuno-compromised patients in diverse clinical settings

Victor H. Ferreira[1,5], Javier T. Solera [1,5], Queenie Hu[2], Victoria G. Hall[1], Berta G. Arbol [1], W. Rod Hardy[2], Reuben Samson[2], Tina Marinelli[1,3], Matthew Ierullo [1], Avneet Kaur Virk[1], Alexandra Kurtesi[2], Faranak Mavandadnejad[1], Beata Majchrzak-Kita[1], Vathany Kulasingam[1], Anne-Claude Gingras [2,4], Deepali Kumar [1,6] & Atul Humar [1,6] ✉

Immunocompromised patients are predisposed to severe COVID-19. Here we compare homotypic and heterotypic humoral and cellular immune responses to Omicron BA.1 in organ transplant patients across a diverse clinical spectrum. We perform variant-specific pseudovirus neutralization assays for D614G, and Omicron-BA.1, -BA.2, and Delta variants. We also measure poly-and monofunctional T-cell responses to BA.1 and ancestral SARS-CoV-2 peptide pools. We identify that partially or fully-vaccinated transplant recipients after infection with Omicron BA.1 have the greatest BA.1 neutralizing antibody and BA.1-specific polyfunctional CD4$^+$ and CD8$^+$ T-cell responses, with potent cross-neutralization against BA.2. In these patients, the magnitude of the BA.1-directed response is comparable to immunocompetent triple-vaccinated controls. A subset of patients with pre-Omicron infection have heterotypic responses to BA.1 and BA.2, whereas uninfected transplant patients with three doses of vaccine demonstrate the weakest comparative responses. These results have implications for risk of infection, re-infection, and disease severity among immune compromised hosts with Omicron infection.

Immunocompromised persons represent a substantial proportion of the general population and are uniquely predisposed to severe sequelae from severe acute respiratory syndrome Coronavirus-2 (SARS-CoV-2) infection[1]. Recipients of an organ transplant are typically treated with life-long exogenous immunosuppression that may impair both cellular and humoral immune responses to common infectious pathogens. Coronavirus disease 2019 (COVID-19) has been noted to result in significant morbidity and mortality in organ

transplant recipients[2,3]. In addition, common preventive strategies including vaccine boosters are less immunogenic[4,5]. The replication permissive host-environment in immunocompromised patients may also serve to promote the evolution of variants underscoring the importance of understanding the immune response to SARS-CoV-2 in these patients[6]. The rapid emergence of the Omicron-BA.1 variant has been especially challenging for immunocompromised patients. For example, we have shown that three doses of mRNA vaccine results in

[1]Department of Medicine, University Health Network, Toronto, ON, Canada. [2]Lunenfeld-Tanenbaum Research Institute, Mount Sinai Hospital, Sinai Health, Toronto, ON, Canada. [3]Department of Infectious Diseases and Microbiology, Royal Prince Alfred Hospital, Sydney, NSW, Australia. [4]Department of Molecular Genetics, University of Toronto, Toronto, ON, Canada. [5]These authors contributed equally: Victor H. Ferreira, Javier T. Solera. [6]These authors jointly supervised this work: Deepali Kumar, Atul Humar. ✉e-mail: atul.humar@uhn.ca

Delta and Omicron neutralization positivity in only 55% and 18% of transplant recipients, respectively[7,8], significantly lower than that reported in the general population. More recently, a subvariant of Omicron, dubbed BA.2, has supplanted BA.1 as the dominant circulating variant within many communities.

Transplant recipients generally also have significantly impaired T-cell responses to pathogens and vaccines[9–13]. This has been observed with two-dose mRNA vaccine strategies and with natural SARS-CoV-2 infection in these patients[14,15]. Although mutations in the Omicron-BA.1 variant significantly reduce the neutralizing capacity of sera compared to earlier variants or ancestral SARS-CoV-2[16,17], cellular immunity studies in immunocompetent individuals suggest that T-cell responses are better conserved, such that 70–80% of the CD4+ and CD8+ T-cell response to ancestral SARS-CoV-2 may be maintained among those who are vaccinated or previously infected with earlier variants (e.g., Alpha, Delta) or ancestral SARS-CoV-2[18]. Whether such heterotypic responses are maintained in immunocompromised hosts remains unknown.

In the current study, we evaluated and compared homotypic and heterotypic humoral and cellular responses to Omicron BA.1 in organ transplant recipients across a set of diverse clinical settings. These included transplant patients recovered from (mostly vaccine breakthrough) infection during the Omicron BA.1 wave, patients infected with SARS-CoV-2 in the pre-Omicron era (e.g., D614G, Alpha or Delta), and uninfected patients receiving three doses of mRNA vaccine. A cohort of immunocompetent, triple-vaccinated healthcare workers with no history of SARS-CoV-2 infection was used for comparison in this study.

## Results
### Study cohorts
Four separate cohorts were analyzed for a total of 246 patients included in the study (Fig. 1). The first cohort consisted of organ transplant patients who had SARS-CoV-2 infection from March 2020 to September 2021 prior to the emergence of the Omicron variant ($n = 91$). Of these, the majority were infected with ancestral SARS-CoV-2 ($n = 71$) and others had infection with Alpha ($n = 17$), Delta ($n = 2$), or unknown ($n = 1$). In this cohort, 72 patients (79.1%) were unvaccinated, 5 (5.5%) received a single dose, and 14 (15.4%) received two doses of vaccine prior to infection. Cohort 2 consisted of 75 transplant patients who were infected during the Omicron-BA.1 wave. Omicron BA.1 represented >98% of circulating cases at the time of infection[19]. Of the recruited patients in cohort 2, typing was available in 20 patients and confirmed Omicron-BA.1 infection in all cases. Most of this cohort was vaccinated, representing breakthrough Omicron-BA.1 infection: 3 (4.0%) were unvaccinated, 2 (2.7%) received a single dose, 12 (16.0%) were double-vaccinated, 53 (70.7%) were triple vaccinated, and 5 (6.7%) had received

four doses of vaccine. All vaccinations occurred prior to infection. Cohort 3 consisted of 60 transplant recipients who received three doses of mRNA-1273 vaccine and had never been infected with SARS-CoV-2, based on the absence of anti-nucleocapsid antibody (Supplementary Fig. 2) and absence of any prior positive viral detection test. Details related to the antibody response in this cohort have been previously reported[4,7,8] but here we add BA.2 responses, and BA.1-specific T-cell data across cohorts. Cohort 4 consisted of immunocompetent healthcare workers who had received three doses of mRNA vaccine BNT162b2 (Pfizer–BioNTech) and had no prior SARS-CoV-2 infection.

Of the 226 organ transplant recipients, the types of transplants included: 98 kidney transplants (43.4%), 43 lungs (19%), 33 livers (14.6%), 28 kidney–pancreas (12.4%), 19 hearts (8.4%), and five kidney–liver combined transplants (2.2%). Demographics and baseline characteristics of the different cohorts including information about vaccination, organ transplanted, and immunosuppressive therapy, are shown in Table 1. Apart from the vaccination status, cohorts 1, 2, and 3 were different in age (cohort 3 patients were significantly older); however, type of transplant and immunosuppressive regimen were similar among the three cohorts (Supplementary Table 1). No patient received B-cell depleting therapies (e.g., Rituximab) as part of their immunosuppressive regimen.

### Transplant recipients infected with ancestral/Alpha/Delta SARS-CoV-2 demonstrate lower heterotypic immunity to Omicron variant
Among transplant patients recovered from infection with ancestral/Alpha/Delta SARS-CoV-2 (Cohort 1; $n = 91$), the median ID50 for neutralizing antibodies against D614G was 3.54 log$_{10}$ (interquartile range [IQR], 3.13 to 4.08), and 3.36 log$_{10}$ (IQR, 2.87 to 3.88) against Delta variant. The median neutralizing heterotypic response to Omicron BA.1 in Cohort 1 was 0 log$_{10}$ (IQR, 0 to 2.13) (Fig. 2A; $p < 0.001$ D614G vs. Delta; $p < 0.001$ D614G vs. BA.1). The proportion of patients in this cohort who were positive for neutralizing antibodies against D614G and Delta was significantly higher than that for Omicron BA.1, 87/91 (95.6%), 86/91 (94.5%), and 45/91 (49.5%), respectively ($p < 0.0001$ for D614G vs. BA.1). We also tested for neutralizing heterotypic antibodies against Omicron BA.2 ($n = 91$; Fig. 2B). BA.1 and BA.2 neutralization responses were correlated (Spearman $\rho = 0.52$, $p < 0.001$). However, heterotypic immunity against BA.2 seemed more robust with several patients who were negative for BA.1 showing a positive neutralization against BA.2. A total of 21/46 (45.7%) BA.1-negative patients were positive for BA.2 neutralizing antibodies).

SARS-CoV-2 S-specific CD4+ and CD8+ T-cell responses to ancestral SARS-CoV-2, or to Omicron-BA.1 peptide pools were assessed in a

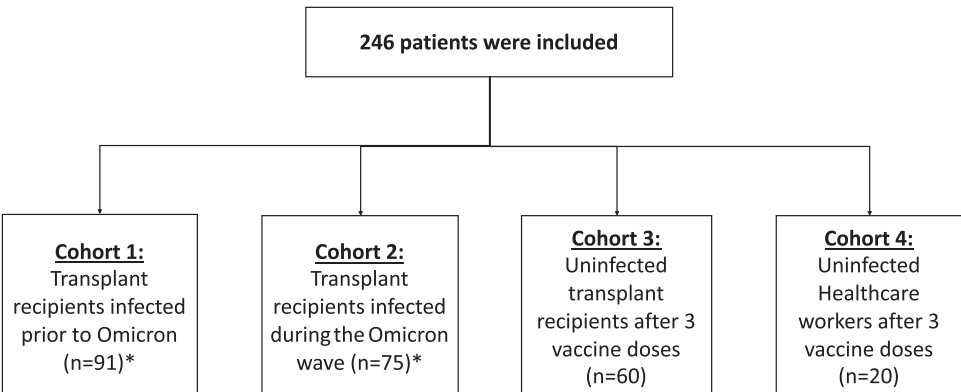

**Fig. 1 | Flow diagram of the study.** A total of 246 participants were included in the study. Cohort 1 ($n = 91$) consisted of transplant recipients infected with ancestral SARS-CoV-2 ($n = 71$), Alpha ($n = 17$), Delta ($n = 2$), or unknown ($n = 1$). Cohort 2 ($n = 75$) includes transplant recipients recovered from natural infection during the Omicron-BA.1 wave. Cohort 3 ($n = 60$) consisted of uninfected transplant patients who received three doses of mRNA-1273 (Moderna) vaccine. Cohort 4 ($n = 20$) was included as a control group and included immunocompetent healthcare workers who received three doses of BNT162b2 (Pfizer–BioNTech) vaccine.

**Table 1 | Demographic and clinical characteristics of the patients of each cohort at baseline**

| Characteristics | Cohort 1 (N = 91) | Cohort 2 (N = 75) | Cohort 3 (N = 60) | Cohort 4 (N = 20) |
|---|---|---|---|---|
| Age—years (mean ± SD) | 54.2 ± 14.1 | 53.5 ± 12.7 | 67.7 ± 4.8 | 45.4 ± 10.7 |
| Female sex—no. (%) | 23 (25%) | 26 (35%) | 23 (38%) | 15 (75%) |
| **Type of transplant—no. (%)** | | | | |
| Kidney | 48 (52.7%) | 30 (40%) | 20 (33.3%) | |
| Lung | 16 (17.6%) | 16 (21.3%) | 11 (18.3%) | |
| Liver | 19 (20.9%) | 11 (14.7%) | 3 (5%) | |
| Heart | 3 (3.3%) | 6 (8%) | 10 (16.7%) | |
| Kidney–pancreas | 4 (4.4%) | 9 (12%) | 15 (25%) | |
| Kidney–liver | 1 (1.1%) | 3 (4%) | 1 (1.67%) | |
| Years since transplant (median, IQR)[a] | 5.4 (2.3–9.1) | 5.9 (2.3–9.9) | 3.8 (2.0–6.7) | |
| **Immunosuppressant—no. (%)** | | | | |
| Prednisone | 73 (80.2%) | 64 (85.3%) | 50 (83.3%) | |
| Tacrolimus | 81 (89%) | 63 (84%) | 47 (78.3%) | |
| Cyclosporine | 3 (3.3%) | 10 (13.3%) | 12 (20%) | |
| Mycophenolate | 65 (71.4%) | 62 (82.7%) | 44 (73.3%) | |
| Azathioprine | 9 (9.9%) | 2 (2.6%) | 7 (11.7%) | |
| Sirolimus | 1 (1.1%) | 1 (1.3%) | 2 (3.3%) | |
| Antilymphocyte globulin last 3 months | 4 (4.4%) | 1 (1.3%) | 0 (0%) | |
| **No. of COVID-19 vaccines—no. (%)** | | | | |
| 0 | 72 (79.1%) | 3 (4%) | 0 (0%) | 0 (0%) |
| 1 | 5 (5.5%) | 2 (2.7%) | 0 (0%) | 0 (0%) |
| 2 | 14 (15.4%) | 12 (16%) | 0 (0%) | 0 (0%) |
| 3 | 0 (0%) | 53 (70.7%) | 60 (100%) | 20 (100%) |
| 4 | 0 (0%) | 5 (6.7%) | 0 (0%) | 0 (0%) |
| **Vaccine brand—no. (%)** | | | | |
| BNT162b2 (Pfizer–BioNTech) | 10 (52.6%) | 35 (48.6%) | 0 (0%) | 20 (100%) |
| mRNA-1273 (Moderna) | 7 (36.8%) | 7 (9.7%) | 60 (100%) | 0 (0%) |
| ChAdOx1-S (AstraZeneca) | 2 (10.5%) | 0 (0%) | 0 (0%) | 0 (0%) |
| Mix | 0 (0%) | 4 (5.6%) | 0 (0%) | 0 (0%) |
| Unknown brand but mRNA vaccine | 0 (0%) | 26 (36.11%) | 0 (0%) | 0 (0%) |
| SARS-CoV-2-related hospitalization | 46 (51%) | 11 (15%) | | |
| SARS-CoV-2 severe disease[b] | 29 (32%) | 6 (8%) | | |

Cohort 1 includes transplant recipients with COVID-19 diagnosis before the Omicron wave. Cohort 2 includes transplant recipients recovered from Omicron-BA.1 infection. Cohort 3 includes uninfected transplant recipients with 3 doses of mRNA-1273 (Moderna) vaccine. Cohort 4 includes uninfected healthcare workers with 3 doses of BNT162b2 (Pfizer–BioNTech) vaccine.
*IQR* Interquartile range, *SD* Standard deviation.
[a]Years from transplant to infection in case of cohort 1 and 2, and from transplant to last dose of the vaccine in case of cohort 3.
[b]SARS-CoV-2 severe disease includes any patient that required supplementary oxygen.

subgroup of 25 patients from this cohort (based on the subset of patients who had cells available for analysis). However, the subset was representative of the overall cohort for factors such as transplant type and immunosuppression (Supplementary Table 2). Homotypic CD4$^+$ T-cell responses to ancestral peptides, including monofunctional and polyfunctional T-cell responses, were detected in most of this cohort (Fig. 2C). The heterotypic CD4$^+$ T-cell response to Omicron-BA.1 peptides in these patients was comparatively lower, mainly with respect to polyfunctional and IL-2 monofunctional CD4$^+$ T cells, with a median

fold reduction of 1.6 ($p = 0.032$) and 5.2 ($p = 0.0069$) relative to the response to ancestral peptides, respectively. Homotypic and heterotypic S-specific CD8$^+$ T cells were detected less frequently in this cohort, with the antigen-specific response primarily defined in terms of monofunctional IFN-γ producing CD8$^+$ T cells (Fig. 2D). Relative to the homotypic CD8$^+$ T-cell response to ancestral SARS-CoV-2 peptides, no significant differences were measured following stimulation with Omicron-BA.1 peptides.

**Organ transplant recipients recovered from infection with Omicron BA.1 show robust immune responses to Omicron BA.1 and BA.2**

We next assessed immune responses in Cohort 2, 75 transplant recipients with Omicron-BA.1 infection, most of who had a vaccine breakthrough infection: 77.3% of patients had three or more doses of vaccine prior to infection (Table 1). We measured the heterotypic and homotypic neutralizing antibody response in these patients. The median ID50 against D614G was 3.79 log$_{10}$ (IQR, 2.89 to 4.28), and the median ID50 against Delta was 3.43 log$_{10}$ (IQR, 2.89 to 3.94). In this cohort, the homotypic neutralizing antibody response directed against Omicron BA.1 was similarly robust, with a median ID50 of 3.35 log$_{10}$ (IQR 2.77 to 3.70), although significantly lower than the median ID50 for D614G (Fig. 3A; $p = 0.0013$ for D614G vs. BA.1). The proportion of patients with a positive homotypic neutralizing response against BA.1 was 66/75 (88%) while the proportion with heterotypic antibody was 66/75 (88%) for D614G and 69/75 (92%) for Delta.

All patients in cohort 2 ($n = 75$) were assessed for neutralizing antibodies against Omicron-BA.2 (Fig. 3B) demonstrating robust heterotypic antibody responses to BA.2 (median ID50 of 3.08, IQR 2.51-3.60). Neutralizing antibodies for BA.2 were well correlated with neutralizing antibodies against BA.1 in this cohort (Fig. 3C, Spearman ρ = 0.62, $p < 0.001$), and all but one patient with positive neutralization against BA.1 demonstrated positive cross-neutralization against BA.2.

Next, we investigated ancestral SARS-CoV-2 and Omicron BA.1-specific T-cell responses in 64 patients from Cohort 2 who had PBMC available (Fig. 3D, E). Robust CD4$^+$ T-cell responses were detected against both ancestral SARS-CoV-2 and BA.1 peptides (Fig. 3D). For example, 32/64 (50.0%) had 100–1000 per 10$^6$ BA.1-specific polyfunctional CD4$^+$ T cells and 27/64 (42.2%) had >1000 per 10$^6$ BA.1-specific polyfunctional CD4$^+$ T cells. Although the frequency of IL-2 monofunctional CD4$^+$ T cells was statistically significantly lower against BA.1 peptides ($p < 0.001$), there was only a 19.7% decrease from ancestral SARS-CoV-2, suggesting that after natural infection of highly immunized transplant patients with Omicron BA.1, homotypic and heterotypic CD4$^+$ T-cell responses to spike peptides are generally similar. CD8$^+$ T-cell responses were also less commonly detected in this cohort, compared to the CD4$^+$ T-cell response (Fig. 3E). No significant differences in magnitude were measured between the CD8$^+$ T-cell response to ancestral SARS-CoV-2 or BA.1 peptides in this cohort.

We looked at demographic factors that were associated with polyfunctional CD4$^+$ or CD8$^+$ responses (Supplementary Tables 3, 4), We limited our analysis to polyfunctional T-cell responses as these are commonly detected in immunogenicity studies of other viral infections and vaccines and are thought to be functionally superior compared to monofunctional cells[20,21]. A positive response was defined by a minimum T-cell frequency of 0.01%, as per previous studies[14]. No baseline factors were found to be significantly associated with more potent responses possibly due to the homogeneity of the exogenous immunosuppression within the group. We also correlated neutralizing antibody titers against BA.1 with polyfunctional CD4$^+$ and CD8$^+$ T-cell responses (Supplementary Fig. S3). BA-1 neutralizing antibody titers did not statistically correlate with polyfunctional CD4$^+$ (Spearman ρ = 0.22, $p = 0.088$) or CD8$^+$ (Spearman ρ = 0.24, $p = 0.056$) T cells, although we did observe a a non-signifcant trend for both T-cell subsets.

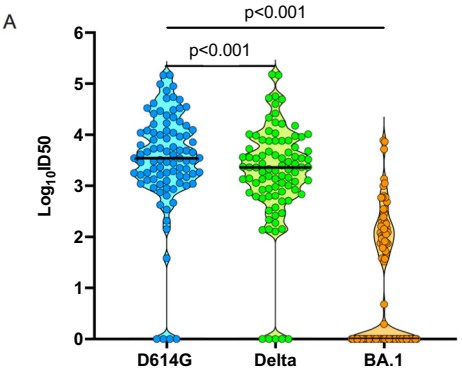

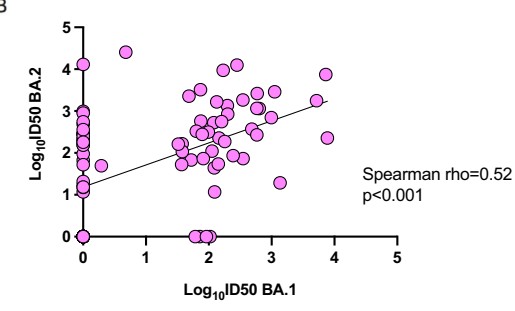

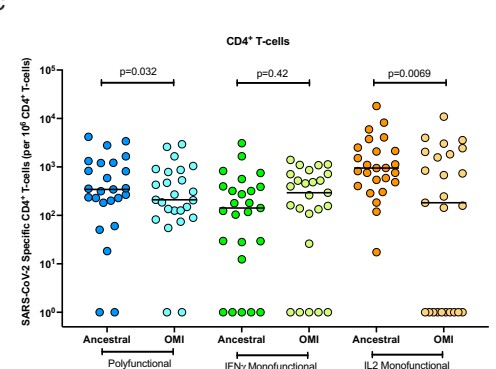

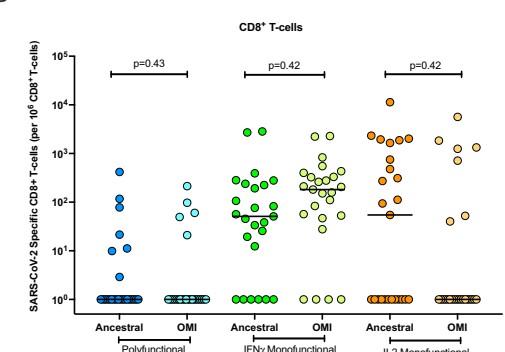

**Fig. 2 | Immune response against ancestral SARS-CoV-2, Delta, and Omicron BA.1 in transplant recipients with non-Omicron SARS-CoV-2 infection (Cohort 1). A** Violin dot plots of the 50% neutralization titers ($\log_{10}$ID50) of neutralizing antibodies against D614G, Delta, and Omicron-BA.1 variants, in transplant recipients infected with non-Omicron variant (primarily ancestral SARS-CoV-2). Each circle represents an individual patient neutralization titer. Horizontal lines represent median values. *P* values were determined using a two-sided Wilcoxon matched-pairs signed-rank test (D614G vs. Delta *p* < 0.001, D614G vs. BA.1 *p* < 0.001). **B** Scatter plot correlation between the $\log_{10}$ID50 of neutralizing antibody titers against Omicron-BA.1 (x-axis) and Omicron-BA.2 (y-axis) in all patients in

Cohort 1 (*n* = 91). A two-tailed Spearman correlation was performed *p* < 0.001). The diagonal line represents the best-fit linear regression line. **C** Proportions of polyfunctional (IFN-γ⁺ and IL-2⁺), IFN-γ monofunctional and IL-2 monofunctional CD4⁺ and **D** CD8⁺ T cells in transplant recipients recovered from non-Omicron variant infection (subset of Cohort 1, *n* = 25). Horizontal lines denote the median for each group. A two-sided Wilcoxon matched-pairs signed-rank test with Holm–Šídák correction for multiple comparisons was used. Adjusted *P* values are shown above each respective comparison. SARS-CoV-2 severe acute respiratory syndrome Coronavirus-2, OMI omicron, IFN-γ interferon gamma, IL-2 interleukin 2. Source data are provided as a Source Data file.

## Breakthrough BA.1 infection induces higher comparative immunity than vaccination alone

We compared neutralizing antibodies directed against BA.1 between the four cohorts (Fig. 4A) and a significant difference in median ID50 titers between groups was found (*p* < 0.001; Kruskal–Wallis test). The neutralizing antibody titer against BA.1 was highest in the transplant cohort (cohort 2) with Omicron-BA.1 infection (Fig. 4A). These titers were even higher than immunocompetent triple-vaccinated healthcare worker controls (median ID50 2.51 $\log_{10}$ (IQR 2.16 to 2.91); *p* < 0.001 vs. BA.1 infected, Fig. 4A). Triple-vaccinated transplant recipients (cohort 3) displayed the lowest neutralizing antibody titers (median ID50 0 $\log_{10}$ [IQR 0-0]; *p* value <0.001 vs. either transplant patients infected with non-Omicron variants (cohort 1) or those with Omicron-BA.1 infection (cohort 2) (Fig. 4A). Proportions of participants that were positive and negative for BA.1 neutralization across the cohorts are shown in Fig. 4B. BA.2 titers in the different cohorts followed a similar trend (Figs. 2B, 3C, Supplementary Fig. S4A, B) with the highest titers being observed in the BA.1 infected cohort.

We next compared polyfunctional CD4⁺ and CD8⁺ T-cell responses to Omicron-BA.1 peptide stimulation in each of the four cohorts, The median frequencies of CD4⁺ polyfunctional T cells were significantly different between groups (Fig. 4C; *p* < 0.001, Kruskal–Wallis test). Results followed a similar trend as seen with BA.1 neutralizing antibodies: the highest BA.1-specific polyfunctional CD4⁺ T-cell response among organ transplant recipients was seen in those

recovered from Omicron-BA.1 infection (cohort 2) (*p* = 0.016 vs. non-Omicron variant infection; *p* < 0.001 vs. triple-vaccinated transplant recipients), with responses comparable to that observed in triple-vaccinated healthcare worker controls (cohort 4) (*p* = 0.22 vs. HCW controls). The next highest response was observed for transplant recipients with non-Omicron variant infection (cohort 1). The lowest responses were observed for the triple-vaccinated, uninfected transplant recipients. Proportions of participants that had low (<100 per 10⁶ cells), medium (100–1000 per 10⁶ cells), and high (>1000 per 10⁶ cells) BA.1-specific CD4⁺ T-cell frequencies are shown in Fig. 4D. Additionally, we compared polyfunctional CD8⁺ T cells between groups (Fig. 4E). Median frequencies between groups were found to be significantly different (*p* < 0.001, Kruskal–Wallis test). Polyfunctional CD8⁺ T-cell responses mirrored both neutralizing antibody and CD4⁺ T-cell responses with the highest polyfunctional CD8⁺ T-cell frequency observed in Cohort 2 (*p* = 0.0002 vs. non-Omicron variant infection; *p* < 0.001 vs. triple-vaccinated transplant recipients), with responses comparable to healthcare worker triple-vaccinated controls (cohort 4) (*p* > 0.99).

## The magnitude of the homotypic immune response following Omicron-BA.1 infection is generally not affected by number of vaccine doses, monoclonal antibody therapy, or disease severity

Since the most robust BA.1 immune response was seen after infection with Omicron-BA.1 (Cohort 2), we sought to determine whether clinical

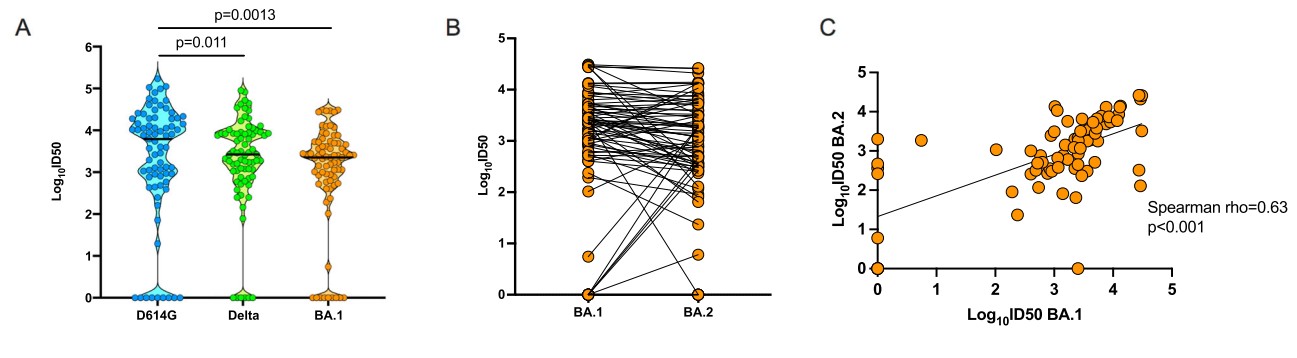

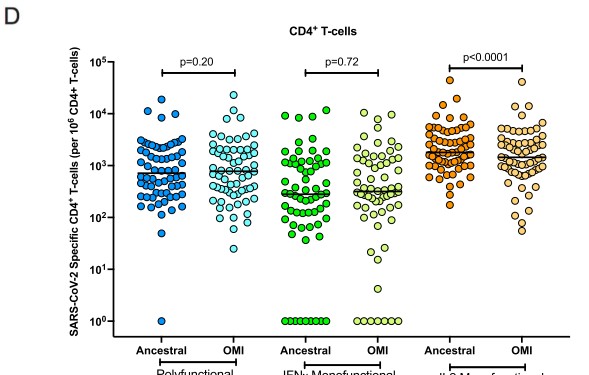

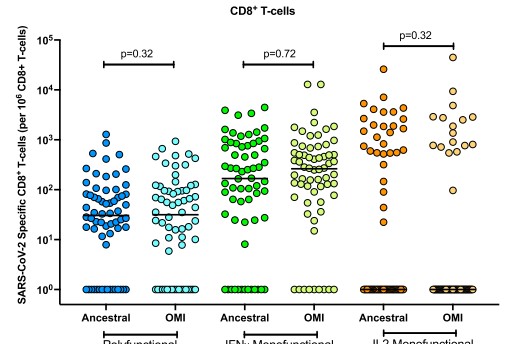

**Fig. 3 | Immune response against ancestral SARS-CoV-2, Delta, and Omicron BA.1 in transplant recipients with Omicron-BA.1 infection. A** Violin dot plots of the 50% neutralization titers (log$_{10}$ID50) of neutralizing antibodies against D614G, Delta, and Omicron BA.1, in transplant recipients post-BA.1 infection (n = 75). Each circle represents an individual patient antibody titer. Horizontal lines represent median values. The *p* value of the difference was estimated using a two-sided Wilcoxon matched-pairs signed-rank test. **B** Paired BA.1 and BA.2 directed neutralizing responses in all patients from the same cohort (n = 75). **C** Scatter plot showing two-tailed Spearman correlation between the log$_{10}$ID50 of neutralizing

antibody titers against Omicron BA.1 (x-axis) and BA.2 (y-axis) in cohort 2 (n = 75; p < 0.001). Diagonal line represents the best-fit linear regression line. Proportions of polyfunctional, IFN-γ monofunctional and IL-2 monofunctional CD4$^+$ T cells (**D**) and **E** CD8$^+$ T cells in a subset (n = 64) transplant recipients with Omicron-BA.1 infection. Horizontal lines denote the median for each group. A two-sided Wilcoxon matched-pairs signed-rank test with Holm–Šídák correction for multiple comparisons was used. Adjusted *P* values are shown above each respective comparison. SARS-CoV-2 severe acute respiratory syndrome Coronavirus-2, OMI omicron, IFN-γ interferon gamma, IL-2 interleukin 2. Source data are provided as a Source Data file.

factors including number of vaccine doses received, treatment with anti-SARS-CoV-2 monoclonal antibody therapy, or disease severity (as measured by oxygen requirement) impacted the magnitude of the homotypic response to BA.1 in this cohort. We compared neutralizing antibody titers to BA.1 in patients with two or fewer doses of vaccine against patients receiving three or more doses of vaccine. This did not significantly affect the convalescent neutralizing antibody titer against BA.1 (Fig. 5A). Similarly, early therapy with SARS-CoV-2 monoclonal antibody (sotrovimab) did not affect the observed homotypic neutralization response against Omicron BA.1 (Fig. 5B). Samples were collected a median of 40 days (IQR 36–44) post-diagnosis suggesting that the half-life of sotrovimab in this setting was not long enough to interfere with assessment of neutralizing antibody. In this cohort, severe COVID-19 occurred in 6/75 (8.0%) patients. The presence of severe COVID-19 did not impact the magnitude of the neutralizing antibody response to Omicron BA.1 (Fig. 5C).

We also looked at whether these clinical factors contributed to the magnitude of the homotypic T-cell response. We compared BA.1-associated CD4$^+$ and CD8$^+$ T-cell responses among those receiving 0–2 doses (n = 13) vs. 3–4 doses (n = 51) of vaccine prior to infection with BA.1. As with the neutralizing antibody response, there was no difference in frequency of BA.1-directed CD4$^+$ or CD8$^+$ T-cell responses between those having 0–2 vs. 3–4 doses of vaccine (p > 0.05; Fig. 5D, E). With respect to use of monoclonal antibody

therapy, no differences were found with respect to use of sotrovimab on T-cell response in this cohort (Fig. 5F, G). An analysis of disease severity was not possible with respect to T-cell responses, due to a low number of patients in the T-cell sub-study requiring oxygen therapy (4/64; 6.3%).

## Anti-RBD antibody levels are correlated with Omicron neutralization

Binding antibodies against the spike protein RBD are generally easier to measure and are often used in clinical settings, although recommendations regarding utility vary. We compared anti-RBD responses in the four cohorts of patients. A Kruskal–Wallis test was performed and median anti-RBD levels were found to be significantly different (p < 0.001). Generally, high anti-RBD antibody levels were observed in all cohorts with the highest responses seen in transplant patients recovered from BA.1 infection and triple-vaccinated healthcare workers (Fig. 6A). Transplant patients infected with non-Omicron variants (unvaccinated) had similar anti-RBD titers to triple-vaccinated transplant recipients (p = 0.236). Across the three transplant cohorts, there was only a modest correlation between anti-RBD level and Omicron-BA.1 neutralizing antibody titer (Fig. 6B–D). However, in the healthcare worker cohort, a strong correlation was observed between anti-RBD level and BA.1 neutralizing antibody titer (Spearman ρ = 0.92, p < 0.001, Fig. 6E).

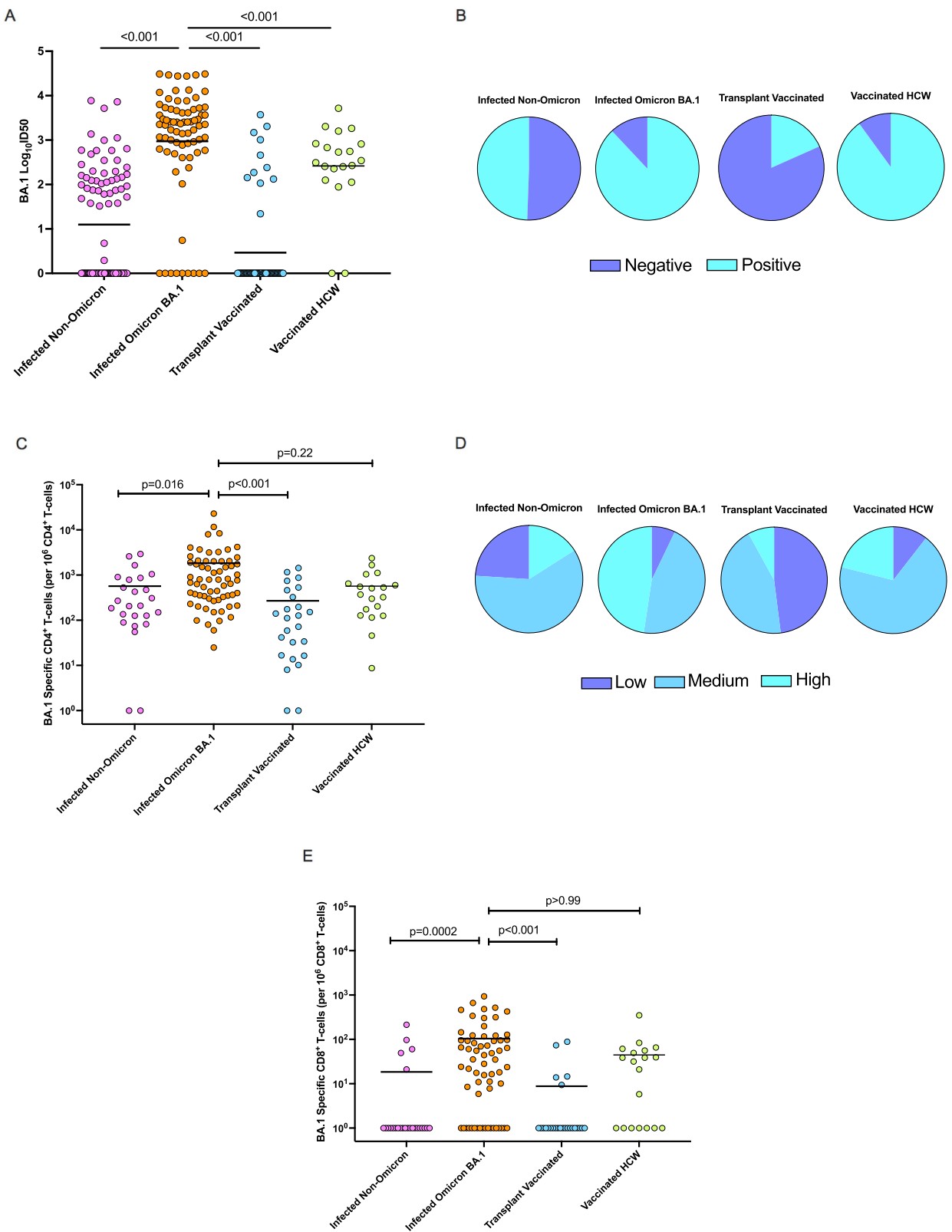

## Discussion

We analyzed humoral and cellular immune responses to SARS-CoV-2 virus with a focus on Omicron-BA.1 variant responses in organ transplant recipients across a diverse spectrum of clinical settings. The major findings of our study can be summarized as follows: The most robust BA.1-directed neutralizing antibody response and BA.1-specific CD4+ T-cell response was observed in the cohort of transplant recipients with Omicron-BA.1 infection, with immune responses comparable to immunocompetent triple-vaccinated controls. Although prior vaccination (ie, hybrid immunity) could have contributed to the high neutralization and T-cell levels, the number of vaccine doses received prior to infection was not associated with the magnitude of response. Triple vaccination alone (without infection) in immunocompromised individuals, was comparatively insufficient for inducing

**Fig. 4 | Comparison of the immune response against Omicron variant between cohorts. A** Dot plots of the Omicron BA.1 50% neutralization titers (log$_{10}$ID50) of neutralizing antibodies across cohorts. Each circle represents an individual participant's neutralization titer. Cohort 1: $n = 91$, Cohort 2: $n = 75$, Cohort 3: $n = 60$, Cohort 4: $n = 20$. Horizontal lines represent median values. A two-sided Kruskal–Wallis test and post hoc Dunn's pairwise test was used. $P$ values are shown above each respective comparison: Cohort 1 vs. 2 $p < 0.001$, Cohort 1 vs. 3 $p < 0.001$, Cohort 2 vs. Cohort 4 $p < 0.001$. **B** Pie-graphs showing proportions of positive and negative BA.1 neutralization for each cohort. **C** Omicron BA.1-specific polyfunctional CD4$^+$ T cells frequencies across groups. Each circle represents a participant's T-cell frequency. Cohort 1 $n = 25$, Cohort 2 $n = 64$, Cohort 3 $n = 25$, Cohort 4 $n = 19$. Horizontal lines denote median values. A two-sided Kruskal–Wallis test with Dunn's

test for multiple comparisons was used. $P$ values are shown above each respective comparison: Cohort 1 vs. 2 $p = 0.016$, Cohort 1 vs. 3 $p < 0.001$, Cohort 2 vs. Cohort 4 $p = 0.22$. **D** Pie-graphs showing proportions of low (<100 per 10$^6$), medium (100–1000 per 10$^6$), and high (>1000 per 10$^6$) BA.1-specific CD4+ T-cell frequencies for each cohort. **E** Omicron BA.1-specific polyfunctional CD8+ T cells frequencies across groups. Each circle represents a participant's T-cell frequency. Cohort 1 $n = 25$, Cohort 2 $n = 64$, Cohort 3 $n = 25$, Cohort 4 $n = 19$. Horizontal lines denote median values. A two-sided Kruskal–Wallis test with Dunn's test for multiple comparisons was used. $P$ values are shown above each respective comparison: Cohort 1 vs. 2 $p < 0.001$, Cohort 1 vs. 3 $p < 0.001$, Cohort 2 vs. Cohort 4 $p > 0.99$. HCW healthcare worker. Source data are provided as a Source Data file.

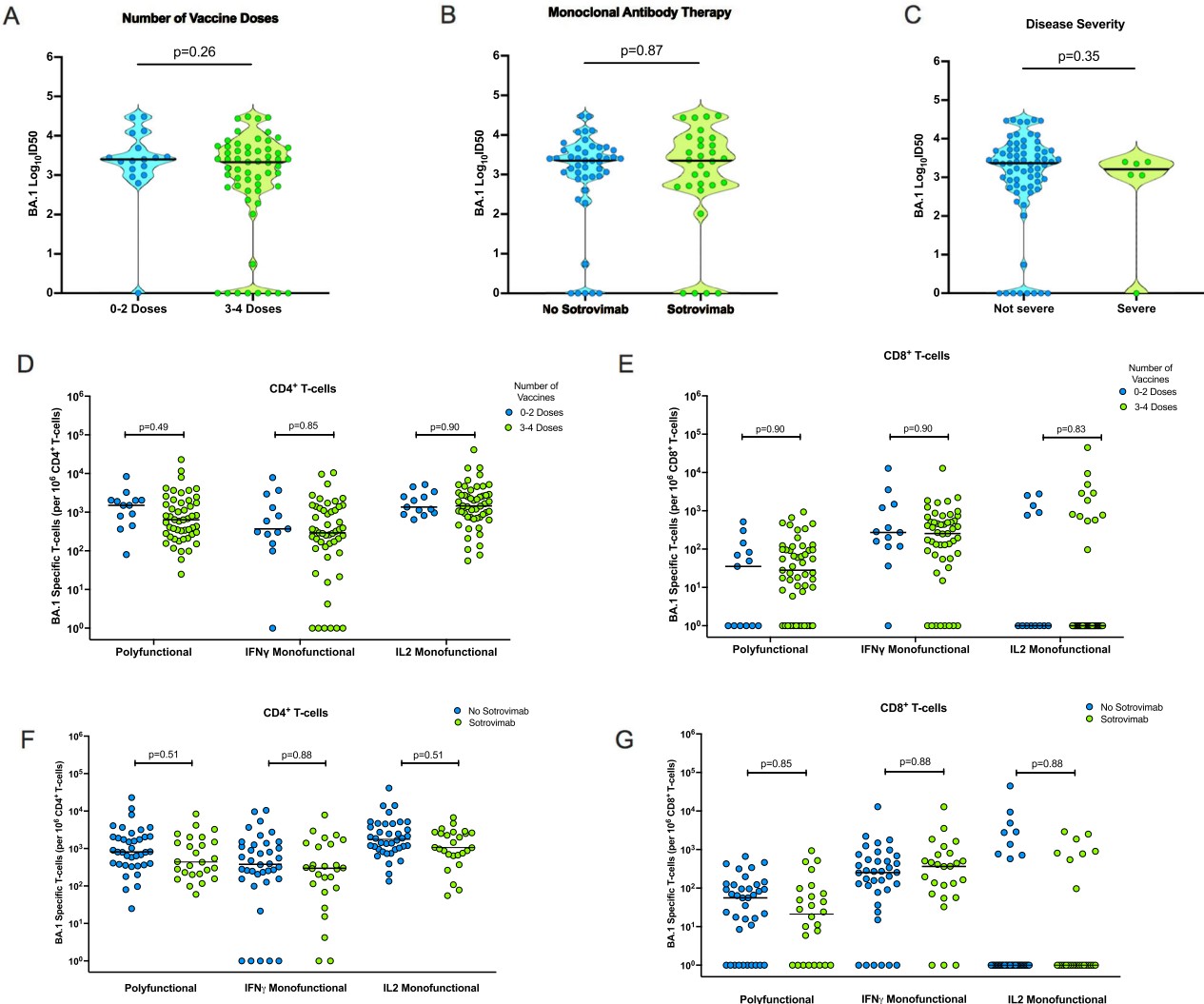

**Fig. 5 | Comparison of the immune response to Omicron BA.1 in transplant patients with BA.1 infection by vaccination status, monoclonal antibody therapy, and severity of disease. A** Violin dot plots of the 50% neutralization titers (log$_{10}$ID50) of BA.1 specific neutralizing antibodies in transplant recipients with Omicron-BA.1 infection, organized according to number of vaccine doses, 0–2 vs. 3–4; (**B**) use of monoclonal antibody therapy (sotrovimab); and **C** severity of disease, defined as requiring oxygen therapy. Horizontal lines represent median values. The $p$ value of the difference between subgroups was estimated using two-

sided Mann–Whitney $U$ test. **D**, **F** Proportions of BA.1-specific polyfunctional, IFN-γ monofunctional and IL-2 monofunctional CD4$^+$ T cells, and **E**, **G** BA.1-specific CD8$^+$ T cells in transplant recipients with Omicron-BA.1 infection ($n = 64$), organized according to number of vaccine doses received prior to infection (**D**, **E**) or use of sotrovimab (**F**, **G**). Horizontal lines denote the median for each group. A two-sided Mann–Whitney $U$ test with Holm–Šídák correction for multiple comparisons was used. Adjusted $P$ values are shown above each respective comparison. IFN-γ interferon gamma, IL-2 interleukin 2. Source data are provided as a Source Data file.

robust BA.1-specific humoral and cellular immune responses. Further, in the Omicron-BA.1 infected group (Cohort 2), a strong cross-reactive neutralizing response was observed against Omicron BA.2, D614G, and Delta, suggesting the capacity for a robust heterotypic neutralizing response. In contrast, the cohort of transplant patients with ancestral

SARS-CoV-2/Alpha/Delta infections (Cohort 1) mounted comparatively poorer heterotypic neutralizing antibody responses to BA.1 (only 40.5% of patients had detectable antibodies) despite having strong neutralizing antibody responses to D614G and Delta. Interestingly, the heterotypic BA.2 responses appeared somewhat better than the

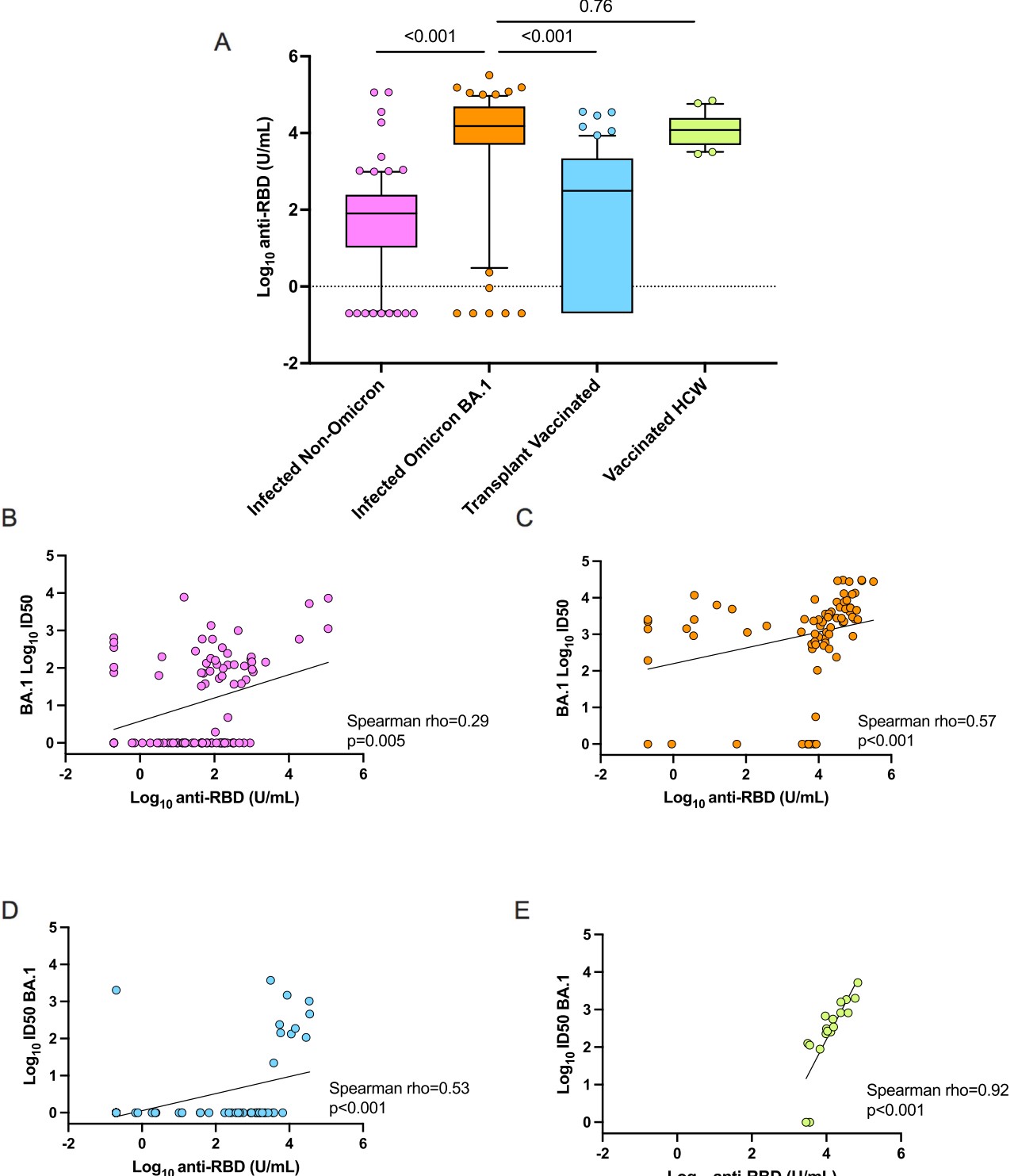

**Fig. 6 | Anti-receptor binding domain (RBD) binding antibody level response by cohort. A** Box and whisker plots of the $\log_{10}$RBD binding antibody titer by cohort. In each box, the horizontal line represents the median value, with the bottom and the top of the box indicating the 25 and 75th percentile, respectively. Cohort 1: $n = 90$, Cohort 2: $n = 75$, Cohort 3: $n = 60$, Cohort 4: $n = 20$. Circles represent outside values (individual patients with antibody titers far from the rest). A two-sided Kruskal–Wallis test and post hoc Dunn's pairwise test was used to compare the BA.1 infected cohort with the other cohorts. *P* values are shown above each respective comparison: Cohort 1 vs. 2 $p < 0.001$, Cohort 1 vs. 3 $p < 0.001$, Cohort 2 vs. 4 $p = 0.76$. **B**–**E** Scatter plots showing two-tailed Spearman correlation between the $\log_{10}$ID50 of neutralizing antibody titers against Omicron BA.1 (y-axis) and the $\log_{10}$ anti-RBD level in U/mL (x-axis) in each cohort: (**B**) infected non-Omicron, $p = 0.005$; (**C**) infected BA.1, $p < 0.001$; (**D**) three-dose vaccinated transplant patients uninfected, $p < 0.001$; and **E** three-dose vaccinated healthcare workers, $p < 0.001$. Diagonal lines represent the best-fit linear regression line. Source data are provided as a Source Data file.

heterotypic BA.1 response in Cohort 1. Neutralizing antibody titers are highly predictive of protection from symptomatic SARS-CoV-2 infection[22] and our findings highlight the susceptibility of this population to currently circulating variants.

Immunosuppression can blunt the response to vaccination and immunocompromised persons are recommended to receive 3 doses of mRNA vaccine as a primary series with a 4th dose booster. The FDA has recently approved a 2nd booster (5th dose) for immunocompromised persons. We found poor responses to BA.1 in 3-dose immunized transplant recipients which may explain the large number of vaccine breakthrough infections in this population. Poor immunogencity of vaccination has been documented in other studies in transplant recipients as well, although fewer data exist for new variants. Charmetant et al. showed that neutralization against ancestral virus (D614G) was improved after 3 doses of mRNA vaccine in kidney transplant patients[23]. Kwon et al. evualated efficacy based on COVID-19 case counts in transplant patients and demonstrated a vaccine effectvenss of 29% with 2 doses of mRNA vaccine and 77% with 3 doses; however this was prior to the emergence of the Omicron variant[24]. Karaba et al.[25] showed two doses of COVID-19 vaccine had poor neutralizing antibody responses against Omicron BA.1, which improved only marginally with subsequent doses. In general, studies have shown that three doses of vaccine results in reasonable neutralization against variants circulating prior to Omicron[7,8,23,25,26]. A reassuring finding is that once partially or fully immunized transplant patients have recovered from infection with Omicron BA.1, homotypic and heterotypic neutralizing antibody titers are strongly boosted, to levels even higher than triple-vaccinated, immunocompetent healthcare workers. Similarly, CD4+ and CD8+ T-cell responses appear to be highly activated, with polyfunctional CD4+ T cells frequencies being increased to magnitudes similar to vaccinated immunocompetent controls. Interestingly the robustness of this response was independent of the number of vaccine doses received, suggesting that natural infection may be the primary trigger for a good response in this population. Omicron-BA.1 infection not only expanded BA.1-directed neutralizing antibody responses, but also heterotypic responses to D614G, Delta and BA.2. In fact, the transplant group with BA.1 infection had a greater heterotypic neutralizing antibody response to D614G (and to a lesser extent the Delta variant) than to BA.1. Similar data was seen in immune-competent populations where vaccinated persons with BA.1 infection developed robust neutralizing antibody titers against BA.2[17]. Although the frequency of IL-2 monofunctional CD4+ T cells was significantly lower to BA.1 peptides ($p < 0.001$), this represented only a 19.7% decrease from ancestral SARS-CoV-2, suggesting that infection with Omicron BA.1 in transplant patients also expands homotypic and heterotypic T-cell responses, in particular, CD4+ T cells, but also CD8+ T cells, to similarly high frequencies. In total, these results provide greater reassurance against symptomatic or severe reinfection with Omicron BA.1, or infection or severe disease with heterotypic variants including ancestral SARS-CoV-2, Delta and BA.2, at least in the short term. A limitation of our study was that we did not have long term follow-up to assess waning of immunity.

Past studies from our group prior to the emergence of Omicron showed that antibody and T-cell responses in organ transplant recipients were higher after infection than vaccination with two doses of mRNA vaccine[14,15]. Here we find further evidence that in transplant patients, natural infection, specifically with the circulating variant, is needed for robust immune responses compared to currently available vaccine strategies (whether an Omicron-specific vaccine would have improved immunogenicity is unknown). This is somewhat in contrast to studies in immune-competent hosts where 3-dose vaccination has been found to induce neutralization that is more comparable to BA.1 infection[17]. We previously saw a similar phenomenon in influenza, where higher T-cell frequencies, and a greater breadth and depth of antibody response was measured following infection with influenza A vs. vaccination[9,27]. The reasons for this are unknown but it may be due

to host factors, including genetics and underlying immunosuppression, that may contribute to prolonged viral replication/shedding[2,28,29] that may in turn facilitate antigen presentation. Further evidence of diverging immune responses between immunocompetent and immunocompromised patients include differences in antibody correlation. In the three transplant cohorts, there was only a modest correlation between anti-RBD level and Omicron BA.1 neutralizing antibody titer, whereas this correlation was much stronger in the healthcare worker cohort (Cohort 4).

Our data shows that heterotypic polyfunctional CD4+ T-cell responses to BA.1 do occur following ancestral SARS-Co2/Alpha/Delta infection (Cohort 1) but are lower than those observed following natural BA.1 infection (Cohort 2), and the latter group mounted robust BA.1 specific T-cell responses. This may have significant clinical implications for immune compromised patients. While neutralizing antibodies appear important for preventing infection and symptomatic disease, it is thought that T cells are critical for minimizing severe COVID-19[30], although meaningful correlates of T-cell protection are still unavailable. In prior studies we found that transplant patients recovered from severe COVID-19 (in the pre-Omicron era) had lower frequencies of antigen-specific T cells at convalescence (as compared with those with milder infection)[14]. Taken together, these data suggest transplant patients with prior immunity obtained via vaccination or prior (non-Omicron) infection, including those with prior critical COVID-19, may continue to be at risk for acquiring a severe Omicron infection.

Our study had some limitations. Neutralization assays were not performed using live SARS-CoV-2 virus assays. However, the use of spike-pseudotyped lentiviral neutralization assays for SARS-CoV-2 immunogenicity are extensively reported in the literature, and the current assay has previously been validated and showed strong correlation to live-virus assays[31]. We acknowledge that the spike-pseudovirus assay and T-cell assays we used do not assess immune responses that may occur to antigens outside the spike protein. Also, while we assessed T-cell immunity specific to BA.1, we did not assess T-cell immune responses specific to BA.2. At the time of this writing, a BA.2 specific peptide pools were not available to us. There was heterogeneity in the time to infection/vaccination post-transplant for all patients, and there were differences in age between cohorts specifically the healthcare worker control group was younger than the transplant cohorts. Despite this, the BA.1 infected cohort showed a more robust immune response. We also acknowledge that under-sampling of severe cases may have occurred as some patients died or were unable to participate in the study due to morbidity from severe disease. Strengths of our study include collection of antibody and T-cell data, the use of well characterized prospectively enrolled clinical cohorts, with detailed data collection, across a diverse clinical spectrum of transplant patients, as well as the comparison to an immunocompetent control group.

In summary, unvaccinated transplant patients with prior SARS-CoV-2 non-Omicron infection, or patients vaccinated with three doses of mRNA vaccine, may be at risk for symptomatic or severe omicron infection, owing to comparatively lower heterotypic antibody and CD4+ T-cell responses to Omicron BA.1. In contrast, partially or fully immunized transplant recipients recovered from Omicron-BA.1 infection developed potent homo- and heterotypic neutralizing antibody and T cells responses (including polyfunctional CD4+ T cells), at magnitudes similar to triple-vaccinated immunocompetent controls. These include cross-reactive responses to BA.2. The risk for reinfection and severe disease may be comparatively mitigated in this cohort.

## Methods

### Study design and ethics
The study was approved by the University Health Network research ethics board. All participants or their delegates provided informed

consent. We recruited four cohorts. The first cohort consisted of organ transplant recipients ($n = 91$) with ancestral SARS-CoV-2 infection, or infection with earlier variants of SARS-CoV-2 that circulated in the collection period, March 2020 to September 2021. The second cohort consisted of transplant recipients with (mostly vaccine breakthrough) Omicron-BA.1 infection, enrolled from December 25, 2021 to January 2022 ($n = 75$). The third cohort consisted of uninfected transplant recipients who received three doses of mRNA-1273 vaccine (Moderna, USA; $n = 60$). The final cohort consisted of a control group of uninfected healthcare workers vaccinated with three doses of BNT162b2 (Pfizer, USA) vaccine ($n = 20$). Infections were documented using SARS-CoV-2 nasopharyngeal swab PCR or rapid antigen tests. Variant determaination of samples was performed using C19-SPAR-Seq[32] in the clincal microbiology laboratory. Peripheral blood mononuclear cells (PBMCs) and serum were collected at 4–12 weeks after symptom onset in infected patients and 4–12 weeks after the third dose of vaccine in the vaccinated cohorts.

### Anti-spike RBD and anti-nucleocapsid binding antibodies

Serologic testing for anti-SARS-CoV-2 spike (S) receptor binding domain (RBD) binding antibodies was performed using the Elecsys SARS-CoV-2 S electrochemiluminescent immunoassay (Roche, Switzerland)[33]. Index measurements ≥0.8 U/mL were considered positive for anti-S antibodies. Antibodies against the nucleocapsid protein of SARS-CoV-2 were measured using a chemiluminescent microparticle immunoassay (Abbott Laboratories, USA) with index measurements of ≥1.4 considered positive as per manufacturer's instructions.

### Variant-specific neutralization assay

To perform the pseudovirus neutralization assay[7], viral packaging (psPAX2; Addgene, USA), the ZsGreen and luciferase reporter (pHAGE-CMV-Luc2-IRES-ZsGreen-W, kindly provided by Jesse Bloom) and the spike protein constructs (D614G, Delta, Omicron-BA.1 and Omicron-BA.2 SARS-CoV-2, generated from consensus sequences in https://outbreak.info) were co-transfected into HEK293T cells (obtained from ATCC, #CRL-3216) using jetPRIME (Polyplus). The viral supernatants were harvested 48 h post transfection, and the viral titre assay for each pseudovirus was performed by infecting HEK293T-ACE2/TMPRSS2 cells (prepared as per Abe et al.[31]), followed by a luciferase assay to determine the relative luciferase unit (RLU). For the neutralization assay, patient serum samples were diluted 1:22.5 and then serially diluted 3-fold over 7 dilutions, followed by incubation with diluted virus at a 1:1 ratio for 1 h at 37 °C prior to addition to HEK293T-ACE2/TMPRSS2 cells. The infected cells were lysed after 48 h using the BrightGlo Luciferase Assay System (Promega, USA), and luciferase activity was measured using a PerkinElmer Envision instrument (PerkinElmer, USA). Both the HEK293T and HEK293T-ACE2/TMPRSS2 cells were maintained at 85% confluency for no more than 25 passages. All cell lines used were tested for, and free of, Mycoplasma contamination.

### T-cell assessment

A total of $10^6$ cryopreserved PBMCs were thawed and rested for 2 h prior to incubation at 37 °C with overlapping peptides (15-mers with 11 amino acid overlaps, including a few 13–17mers) corresponding to the complete SARS-CoV-2 S protein[4,15,34]. Peptides corresponding to ancestral SARS-CoV-2 were purchased from Miltenyi Biotec (Germany) and resuspended in sterile water; Omicron-BA.1 peptides were purchased from JPT (Germany) and resuspended partially in DMSO and then PBS. The sequences used for designing ancestral SARS-CoV-2 and BA.1 peptides were MN908947.3 (GenBank), and EPI_ISL_6752027 (Gisaid), respectively. The mutational profiles associated with these products are available on the manufacturers' websites. After fixation and permeabilization, intracellular cytokine

staining (ICS) was performed using antibodies against interferon-gamma (IFN-γ) and interleukin-2 (IL-2). Negative controls consisted of PBMCs incubated overnight with media alone or media containing an equivalent percent of DMSO (0.2%) as in Omicron peptide-stimulated samples. PMA/ionomycin (ThermoFisher Scientific) was used as a positive control. Frequency of S-specific T cells was determined by subtracting the background cytokine frequency from the frequency in peptide-stimulated samples. Cells expressing IFN-γ and IL-2 alone (monofunctional), or both cytokines simultaneously (polyfunctional) were evaluated in this study. A minimum of 100,000 live $CD3^+$ T cells were required for samples to be included in the flow analysis. Representative gating is shown in Supplementary Fig. S1. The following antibodies were used in the study: anti-human CD3-BV786 (clone SK7, dilution 1:80, cat no: 563799,), anti-human CD4-Pacific Blue (clone RPA-T4, dilution 1:40, cat no: 558116) anti-human CD8-APC-Cy7 (clone RPA-T8, dilution 1:40, cat no: 557760), anti-human IFN-γ-FITC (clone B27, dilution 1:40, cat no: 554700), and anti-human IL-2-APC (clone MQ1-17H12, dilution 1:40, cat no: 500310). Antibodies were purchased from BD Biosciences, except for the anti-IL-2 antibody which was purchased from BioLegend. Validation of antibodies for specificity and application was performed by vendors, and corresponding data are available from suppliers. Flow cytometry data was acquired using BD FacsDiva version 6.1.3 and analyzed using FlowJo version 10.7.1.

### Statistics

Descriptive statistics were used to outline the baseline characteristics of each cohort. ID50 (inhibitory dilution with 50% virus neutralization) titers were calculated in Prism using a nonlinear regression (log[inhibitor] versus normalized response–variable slope) algorithm and converted to a $log_{10}$ scale. A positive neutralization assay can be defined as any dilution that results in 50% viral neutralization as calculated based on the above generated curve (therefore the lower limit of detection is considered $log_{10}ID50 > 0$ where $10^0$ corresponds to absence of 50% neutralization as calculated for an undiluted specimen). Differences in the medians of paired observations were tested using the Wilcoxon matched-pairs signed-rank test, and unpaired data was compared with Mann–Whitney U test (Wilcoxon rank-sum test). Kruskal–Wallis tests were used to test differences in the median between multiple cohorts. Differences in proportions were compared using Chi-square or Fisher's exact test. Spearman rank correlation test was used to measure the degree of correlation between the different antibody titers. Holm-Sidak adjusted significance levels were used to correct for multiple comparisons, where appropriate. $P$ values <0.05 were considered significant. Statistical analyses was performed with Prism version 9.1.1 and 9.2.0 (GraphPad Software, USA) and Stata statistical software, version 15.1 (StataCorp, LLC, College Station, TX, USA). Figure rendering was performed with Prism.

### Reporting summary

Further information on research design is available in the Nature Research Reporting Summary linked to this article.

## Data availability

The patient-specific clinical data that support the findings of this study are available from the corresponding author upon reasonable request with the required ethical approvals. All data generated from patient samples are presented in the paper. Source Data are provided with this paper.

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

## Acknowledgements

The study authors would like to acknowledge all study participants, and Ilona Bahinskaya and Natalia Pinzon for their expert clinical study coordination. We thank CoVaRR-Net, the Coronavirus Variants Rapid Response Network of the Canadian Institutes for Health Research, for the development of the variant-specific lentivirus assays, and Jesse Bloom for the initial lentiviral constructs. A.C.G. is pillar lead for CoVaRR-Net and Canada Research Chair in Functional Proteomics. This work was supported by the Public Health Agency of Canada through the COVID-19 Immunity Task Force and Vaccine Surveillance Reference Group (grant no: 2122-HQ-000067), awarded to D.K., A.H., V.H.F.), in partnership with the Canadian Donation and Transplantation Research Program (CDTRP).

## Author contributions

Conceptualization, A.H., D.K., V.H.F., J.T.S.; methodology, V.H.F., A.C.G., Q.H., W.R.H., R.S., A.K., F.M.; formal analysis, A.H., D.K., V.H.F., J.T.S., Q.H., V.K.; investigation, V.H.F., J.T.S., V.G.H., B.G.A., T.M., M.I., A.K.V., A.K., F.M., B.M.K., V.K.; writing—original draft, A.H., D.K., V.H.F., J.T.S.; writing—review and editing, all authors; visualization, A.H., D.K., V.H.F., J.T.S.; supervision, A.H., D.K., V.H.F., A.C.G., V.K.; funding acquisition, A.H., D.K., V.H.F., V.G.H.

## Competing interests

D.K. has received research grant from Roche, GSK, and advisory fees from Roche, GSK, Sanofi, Merck and Exevir. A.H. has received research grants from Roche and Merck and advisory fees from Merck and Takeda. A.C.G. has received research funds from a research contract with Providence Therapeutics Holdings, Inc for other projects. No other authors have competing interests to disclose.

## Additional information

**Peer review information** *Nature Communications* thanks Alan Koff and other anonymous reviewer(s) to the peer review of this work. Peer review reports are available.

