## [Peer Review File · Nature Communications]

REVIEWER COMMENTS

Reviewer #1 (Remarks to the Author):

The authors present an interesting and timely analysis of humoral and cell-mediated immune responses across several cohorts of solid organ transplant recipients, with comparison to a control group of healthy vaccinated participants. Their findings support the notion of robust heterotypic immunity among (predominantly vaccinated) transplant recipients who have recovered from COVID-19 infection, and re-emphasize the relatively poor responses among patients receiving 3 doses of mRNA vaccine alone. The novel findings of this study include the comparison of neutralization among solid organ transplant recipients recovering from BA.1 infection with extrapolation to protection from BA.2, along with evaluation of expected immune responses to BA.2 among vaccinated transplant recipients or transplant recipients with prior infection from a different variant. These findings further support a need for alternate strategies to standard vaccination, including omicron-specific vaccines or pre-exposure prophylaxis.

The following comments/suggestions are noted below:

Line 70 – Based on my interpretation of your methodology, the determination that a patient had BA.1 infection was that they had COVID infection during BA.1 predominance, and sampling of a subset of the population showed all of the samples were omicron. This was then extrapolated to suggest that all patients infected during omicron prevalence had omicron. If this is the case, I would clarify the wording to account for the fact that cohort two is based on this assumption. Based on high community prevalence I would agree this assumption is likely valid but it is an assumption.

Line 94 – would clarify site of PCR; nasopharyngeal, anterior nares etc.

Line 95 – please spell out acronym PBMC the first time

Line 172 – These numbers do not add up to 100%, please clarify

Line 176 – remove word 'and' which appears twice

Line 180 – Nucleocapsid antibody titers may wane over time and rarely could become negative. I would define cohort 3 as never being infected with SARS-CoV-2 based on the absence of a positive nucleocapsid antibody (and presumably absence of prior positive test).

Line 184 – how did you define healthy?

Line 195 – would clarify for reader this is referring to Cohort 1, and in subsequent sections would list which cohort you are discussing for easier comparison to figures

Line 202 – I find this wording confusing. To clarify, this is still referring to Cohort 1 patients who were never exposed to omicron, and the “BA.1” positives were those positive for heterotopic neutralizing antibodies? If so please consider revising for reader clarification.

Lines 208, 231, 237 – how were these subgroups chosen?

Line 247/Figure 3E – While not statistically significant, there appears to be a clear trend and it might still be worth mentioning the difference between IL-2 monofunctional CD8+ T-cell responses between variants, a trend also noted in the CD4+ graph

Line 258 – I do not think Figure 2B would be strong enough correlation to consider a trend

Line 277 – Would specify the cohort you are referring to

Line 282 – Would administration of monoclonal antibodies alter the findings of your subsequent neutralization assays given the long half life? If so this may affect the neutralizing antibody component of those recovered from omicron natural infection

Line 324 – please review the phrasing of this sentence

Line 339 – consider changing the word fulsome

Line 356 – I would clarify that natural infection is generally needed for robust immune responses compared with currently available vaccination strategies. I would also clarify that natural infection with a circulating variant may be needed rather than natural infection alone. Whether an omicron specific vaccine would mirror the findings with natural infection is not known.

Line 374 – I would clarify the comparison group when you mentioned “lower frequencies”.

Figure 1 - The figure makes it seem as though Cohort 1 patients were vaccine naïve, would consider revising, as the text notes some were vaccinated.

Figure 6A – Can you please explain the negative log anti-RBD measurements

Table 1 – Please ensure # of patients in cohort 1 COVID vaccines column adds to n=91

Table 1 – Review the numbers and percentages for cohort 1 vaccine brand

Table 1 – Recent treatment for rejection, B-cell depleting therapies would be helpful as additional demographic criteria of interest

-A limitation of this study is that it does not account for patients in cohort 2 who died or were potentially unable to participate in the study due to morbidity from severe disease. This may skew the immune response findings towards the higher end. As you noted in your citation, those with severe diseases had poorer T-cell responses (citation 14).

Reviewer #2 (Remarks to the Author):

The authors present a retrospective observational study on the humoral and cellular immune response to SARS-CoV-2 variants in a heterogeneous group of solid organ transplant recipients (SOTR) who differ by the number of vaccine doses received and presence or absence of breakthrough infection in addition to vaccination. They include a small group of non-immunosuppressed controls as a comparator. They measure anti-RBD antibodies, neutralizing antibodies using pseudovirus, and spike specific polyfunctional CD4 and CD8 T cells using peptide stimulation and intracellular cytokine staining with flow-cytometry. They report that SOTRs who are at least partially vaccinated and then infected with BA.1 develop more robust antibody and T cell responses both to BA.1, but other variants (specifically Delta) than SOTRs who were infected pre-omicron (and largely unvaccinated) or SOTRs who received three doses of vaccine. These findings are mostly supported by their data and not terribly surprising given what is already known about hybrid (infection plus vaccination) immunity in the non-immunocompromised population. What is perhaps surprising, is that the vaccinated plus infected SOTRs have higher antibody titers against BA.1 than vaccinated (but uninfected) healthy controls. Strengths of the manuscript include the measurement of antibody and T cell responses and the inclusion of these special populations of immunosuppressed patients with varying levels of antigen exposure. Investigation of heterotypic immune responses in SOTRs with hybrid immunity is certainly of interest to the field of transplantation medicine and infectious diseases, and their data provide evidence that it is possible to get SOTRs (traditionally poor responders to SARS-CoV-2 vaccines) to develop immune responses to SARS-CoV-2 (though I don't think anyone would argue that breakthrough infection is the ideal strategy). The somewhat significant limitation is that not all subjects are included in all assays (particularly the T cell assays) so it is difficult to draw strong conclusions in this specific cohort let-alone the broader transplant population based on these data. This could be perhaps overlooked if the authors explored mechanism(s) for their findings, but the study is largely descriptive in nature.

Major Concerns:

1. There is not enough information provided about the various subsets of the cohorts studied in the different assays to draw conclusions. Why were only a small fraction tested against BA.2 for neutralization? Given the current variant climate, heterotypic responses to BA.2 are much more interesting than Delta. The author's do not provide enough detail on the subset of patients that were tested against BA.2 nor do they provide details about the 42 patients from the omicron infected cohort. Therefore, the reader cannot evaluate how representative these subjects are of the larger cohort. Perhaps these 42 were the only ones with adequate cell numbers to perform the assay, then the conclusion that infection plus vaccination leads to better responses would ignore subjects that are so lymphopenic that one cannot measure responses.
2. Cohort 1 is not a proper comparator. Cohort 1 is made up of mostly unvaccinated SOTRs infected pre-omicron. Therefore, they differ from cohorts 2 and 3 by two variables (infection with a different variant

and lack of vaccination). There are already published data that show that infection alone with non-omicron virus does not lead to protection from omicron (10.1056/NEJMc2200133) in healthy controls. Therefore, the data from cohort 1 do not contribute significantly to the message in this manuscript.

3. There are missed opportunities to explore correlations between T cell and antibody responses. Were there subjects that made poor antibody responses, but had good T cell responses? Are these two aspects of the immune response linked as tightly in this immunocompromised population as previously thought? What about the outliers that failed to make T cell responses? Are there specific demographic or clinical factors that explain this lack of response?

Minor concerns:

-time from events are very heterogenous, this should be addressed or controlled for in some way

-what is the source of the 293 cells and the 293-ACE2/TMPRSS2 cells?

-lack of consistency regarding the naming of the variants (strains vs. variants), not sure which is correct, but should be consistent

-additional current literature on this topic should be cited and the current study put in context of these other studies (not all need to be included, but some suggestions): 10.1016/j.jhep.2022.03.042, 10.1097/TP.0000000000004140, 10.1001/jamanetworkopen.2022.6822, <https://doi.org/10.1093/infdis/jiac118>, 10.1126/scitranslmed.abl6141

-the term “wild-type” virus is imprecise and implies these variants were somehow modified rationally. Is this D614G? Wuhan-1? WA-1? Vaccine spike? Would replace the term wild-type with something more specific

-there is no such thing as a “COVID-19 infection.” It is a SARS-CoV-2 infection.

-data on how the authors determined what variant infected each patient is not clear. Were the viral genomes sequenced? There are no methods on this.

-typo in the sentence “Of the recruited patients, typing was available in 20 patients and and confirmed Omicron BA.1 infection in all cases.”

-where are the stats comparing the demographic and clinical aspects of the various cohorts in Table 1? To that end, it appears cohort 3 is on average closer to transplant? Could that partially explain the poorer response?

-In Figure 2a where is the LLOD indicated?, shouldn't this be Log3 transformed given the authors performed serial 3-fold dilutions (Figure 3A too)

-figure 2b is not compelling. Why only test 10 samples? It seems like there are a couple of outliers that are really skewing things

-how were the 25 subjects in figure 1c and figure 1d chosen? How representative of cohort 1 are they (see major concerns above)?

-is there anything special about the people that DID have CD8 positive T cells? Non-kidney? Immunocompromised regimen (see major concerns above)?

- the anti-N data demonstrating that patients weren't infected should be shown.
- in figure 4a a Kruskal-wallis test before testing individual group comparisons would be appropriate
- in figure 4c, are these all the data?
- where are the CD8s in Figure 4?
- In Figure 5B when were these data collected relative to infection (and possible mAb treatment)? Are you actually measuring sotrovimab in these plasma samples?
- Figure 6a needs Kruskal-wallis test
- Figure 6 B – E should have the same X-axis scale
- conclusion about vaccine + infection (non-omicron) being insufficient is not supported by the data because cohort 1 is largely unvaccinated

Reviewer #1 (Remarks to the Author):

The authors present an interesting and timely analysis of humoral and cell-mediated immune responses across several cohorts of solid organ transplant recipients, with comparison to a control group of healthy vaccinated participants. Their findings support the notion of robust heterotypic immunity among (predominantly vaccinated) transplant recipients who have recovered from COVID-19 infection and re-emphasize the relatively poor responses among patients receiving 3 doses of mRNA vaccine alone. The novel findings of this study include the comparison of neutralization among solid organ transplant recipients recovering from BA.1 infection with extrapolation to protection from BA.2, along with evaluation of expected immune responses to BA.2 among vaccinated transplant recipients or transplant recipients with prior infection from a different variant. These findings further support a need for alternate strategies to standard vaccination, including omicron-specific vaccines or pre-exposure prophylaxis.

Response: We thank the reviewer for these comments.

The following comments/suggestions are noted below:

Line 70 – Based on my interpretation of your methodology, the determination that a patient had BA.1 infection was that they had COVID infection during BA.1 predominance, and sampling of a subset of the population showed all of the samples were omicron. This was then extrapolated to suggest that all patients infected during omicron prevalence had omicron. If this is the case, I would clarify the wording to account for the fact that cohort two is based on this assumption. Based on high community prevalence I would agree this assumption is likely valid but it is an assumption.

Response: The reviewer's interpretation is correct. We have clarified the wording. Please see line 71.

Line 94 – would clarify site of PCR; nasopharyngeal, anterior nares etc.

Response: These were nasopharyngeal swabs. We added this to Line 96.

Line 95 – please spell out acronym PBMC the first time

Response: As suggest, we have now spelled out the acronym. Please see Line 98.

Line 172 – These numbers do not add up to 100%, please clarify

Response: We thank the reviewer for bringing this error to our attention. We have corrected these numbers. Please refer to lines 178.

Line 176 – remove word ‘and’ which appears twice

Response: The duplication has been removed.

Line 180 – Nucleocapsid antibody titers may wane over time and rarely could become negative. I would define cohort 3 as never being infected with SARS-CoV-2 based on the absence of a positive nucleocapsid antibody (and presumably absence of prior positive test).

Response: We thank the reviewer for this suggestion. The changes in accordance are found on lines 188-9.

Line 184 – how did you define healthy?

Response: These were health care workers who were not on any exogenous immunosuppression and who did not have any underlying immunodeficiency. We have removed the term “healthy” and simplified to ‘immunocompetent health care workers’. Line 191

Line 195 – would clarify for reader this is referring to Cohort 1, and in subsequent sections would list which cohort you are discussing for easier comparison to figures

Response: We reviewed the manuscript text and clarified throughout which cohorts were being discussed, as per the reviewer’s suggestion

Line 202 – I find this wording confusing. To clarify, this is still referring to Cohort 1 patients who were never exposed to omicron, and the “BA.1” positives were those positive for heterotopic neutralizing antibodies? If so please consider revising for reader clarification.

Response: Yes, the reviewer is correct. As per the previous comment, we provided additional clarification when referring to specific cohorts. We hope this will provide additional clarity.

Lines 208, 231, 237 – how were these subgroups chosen?

Response: Subgroups were selected randomly. However, to eliminate bias in selection, and to address comments from reviewer 2, we have newly performed BA.2 neutralizing antibody testing on all participants. We believe this significantly strengthens the manuscript.

Line 247/Figure 3E – While not statistically significant, there appears to be a clear trend and it might still be worth mentioning the difference between IL-2 monofunctional CD8+ T-cell responses between variants, a trend also noted in the CD4+ graph

Response: We were intrigued by this pattern, too. However, after testing additional samples, the trend toward higher frequencies of IL-2 monofunctional CD8+ T-cells was no longer apparent.

Line 258 – I do not think Figure 2B would be strong enough correlation to consider a trend

Response: We performed additional BA.2 testing of all samples now in order to properly address heterologous responses against BA.2 with a more robust sample size. An updated correlation between BA.1 and BA.2 for all participants can now be found in Figure 2B (for cohort 1) and other figures for the remaining cohorts.

Line 277 – Would specify the cohort you are referring to

Response: We have clarified throughout this paragraph the cohorts and similarly throughout the manuscript. (Line 278-283)

Line 282 – Would administration of monoclonal antibodies alter the findings of your subsequent neutralization assays given the long half life? If so this may affect the neutralizing antibody component of those recovered from omicron natural infection

Response: This is a good point. In order to try and assess this possibility, we looked at the impact of Sotrovimab vs. no-sotrovimab on the detection of neutralizing antibody in this cohort. We did not see any significant difference (Figure 5B) suggesting that Sotrovimab itself did not have a major influence on the subsequent level of neutralization detection. The median time from COVID-19 diagnosis to sample collection was 40 days (IQR 36-46) and patients typically received Sotrovimab 1-2 days following diagnosis. Therefore significant time did elapse between Sotrovimab and sample collection; it may be possible that in the presence of illness the half-life of sotrovimab may be too short to allow detection this long after administration. This point has been added to the results section – see Lines 317-319.

Line 339 – consider changing the word fulsome

Response: We replaced the word “fulsome” and reworded this sentence. Please see Line 388.

Line 356 – I would clarify that natural infection is generally needed for robust immune responses compared with currently available vaccination strategies. I would also clarify that natural infection with a circulating variant may be needed rather than natural infection alone. Whether an omicron specific vaccine would mirror the findings with natural infection is not known.

Response: We thank the reviewer for these suggestions. The text in the relevant section has been modified to reflect these points. Please refer to lines 405-407.

Line 374 – I would clarify the comparison group when you mentioned “lower frequencies”.

Response: This has been clarified, please refer to lines 427 (as compared to those with milder infection).

Figure 1 - The figure makes it seem as though Cohort 1 patients were vaccine naïve, would consider revising, as the text notes some were vaccinated.

Response: The figure has been revised to better reflect the relative vaccination status of each cohort. Please see Figure 1.

Figure 6A – Can you please explain the negative log anti-RBD measurements

Response: Anti-RBD antibody concentrations were log₁₀ transformed in order to show on the figure. The threshold value for a positive anti-RBD with this assay is 0.8 U/mL. Therefore since some values are < 1.0 U/mL, this corresponds to a negative log₁₀. For example, the limit for positivity, 0.8 U/mL, corresponds to a log₁₀ value of -0.97.

Table 1 – Please ensure # of patients in cohort 1 COVID vaccines column adds to n=91

Response: We thank the reviewer for bringing this to our attention. We have now corrected the numbers with respect to number of vaccine doses received (Cohort 1). The correct numbers can now be found in Table 1

Table 1 – Review the numbers and percentages for cohort 1 vaccine brand

Response: We thank the reviewers for also bringing this to our attention. We reviewed the data carefully and provide corrected numbers in Table 1.

Table 1 – Recent treatment for rejection, B-cell depleting therapies would be helpful as additional demographic criteria of interest

Response: We thank the reviewers for considering this. We have added in recent acute rejection in the preceding 3-months and recent T-cell depletion (thymoglobulin) to Table 1 and Supplementary Table 1. In Canada, B-cell depleting therapy (e.g. Rituximab) is not approved for induction or treatment of rejection in solid organ transplant recipients and so is uncommonly used at our center except for the treatment of post-transplant lymphomas. Therefore none of the patients in the current study had received B-cell depleting antibody therapy in the preceding 3 months.

-A limitation of this study is that it does not account for patients in cohort 2 who died or were potentially unable to participate in the study due to morbidity from severe disease. This may skew the immune response findings towards the higher end. As you noted in your citation, those with severe diseases had poorer T-cell responses (citation 14).

Response: We looked at whether severity of disease impacted antibody responses against BA.1 in Cohort 2 and did not find a relationship between magnitude of antibody response and the need for oxygen supplementation, or other higher level of care (Figure 5C). However, the number of patients with severe outcomes in our analysis was low – particularly for the T-cell response - and it's possible that inclusion of patients with severe outcomes like ICU admission and death, would skewer responses lower. We agree with the reviewer about the potential for under-sampling of severe cases and have added this as a limitation to the Discussion section. Please see Lines 442-443.

Reviewer #2 (Remarks to the Author):

The authors present a retrospective observational study on the humoral and cellular immune response to SARS-CoV-2 variants in a heterogeneous group of solid organ transplant recipients (SOTR) who differ by the number of vaccine doses received and presence or absence of breakthrough infection in addition to vaccination. They include a small group of non-immunosuppressed controls as a comparator. They measure anti-RBD antibodies, neutralizing antibodies using pseudovirus, and spike specific polyfunctional CD4 and CD8 T cells using peptide stimulation and intracellular cytokine staining with flow-cytometry. They report that SOTRs who are at least partially vaccinated and then infected with BA.1 develop more robust antibody and T cell responses both to BA.1, but other variants (specifically Delta) than SOTRs who were infected pre-omicron (and largely unvaccinated) or SOTRs who received three doses of vaccine. These findings are mostly supported by their data and not terribly surprising given what is already known about hybrid (infection plus vaccination) immunity in the non-immunocompromised population. What is perhaps surprising, is that the vaccinated plus infected SOTRs have higher antibody titers against BA.1 than vaccinated (but uninfected) healthy controls. Strengths of the manuscript include the measurement of antibody and T cell responses and the inclusion of these special populations of immunosuppressed patients with

varying levels of antigen exposure. Investigation of heterotypic immune responses in SOTRs with hybrid immunity is certainly of interest to the field of transplantation medicine and infectious diseases, and their data provide evidence that it is possible to get SOTRs (traditionally poor responders to SARS-CoV-2 vaccines) to develop immune responses to SARS-CoV-2 (though I don't think anyone would argue that breakthrough infection is the ideal strategy). The somewhat significant limitation is that not all subjects are included in all assays (particularly the T cell assays) so it is difficult to draw strong conclusions in this specific cohort let-alone the broader transplant population based on these data. This could be perhaps overlooked if the authors explored mechanism(s) for their findings, but the study is largely descriptive in nature.

Response: We thank the reviewer for this thorough review. Below are point by point responses and in particular we have performed more testing to help alleviate some of the major concerns.

Major Concerns:

1. There is not enough information provided about the various subsets of the cohorts studied in the different assays to draw conclusions. Why were only a small fraction tested against BA.2 for neutralization? Given the current variant climate, heterotypic responses to BA.2 are much more interesting than Delta. The author's do not provide enough detail on the subset of patients that were tested against BA.2 nor do they provide details about the 42 patients from the omicron infected cohort. Therefore, the reader cannot evaluate how representative these subjects are of the larger cohort. Perhaps these 42 were the only ones with adequate cell numbers to perform the assay, then the conclusion that infection plus vaccination leads to better responses would ignore subjects that are so lymphopenic that one cannot measure responses.

Response: We thank the reviewer for these comments.

We agree this was a major limitation of the previous version. Therefore, to address the potential bias in subset selection, we have now performed additional BA.2 neutralizing antibody testing on all participants in all four cohorts. We agree that heterotypic responses against BA.2 are significantly more relevant than those directed at supplanted variants, like Delta, and therefore have added this additional data to the revision.

In terms of the 42 omicron infected patients we testing for T-cell responses, we have now gone ahead and tested the remaining patients who supplied PBMCs. We now show results for the majority of this cohort (64/75 of the patients) and we believe this strengthens the conclusion and avoids sampling bias. We thank the reviewer for encouraging us to carry out these additional experiments.

2. Cohort 1 is not a proper comparator. Cohort 1 is made up of mostly unvaccinated SOTRs infected pre-omicron. Therefore, they differ from cohorts 2 and 3 by two variables (infection with a different variant and lack of vaccination). There are already published data that show that infection alone with non-omicron virus does not lead to protection from omicron (10.1056/NEJMc2200133) in healthy controls. Therefore, the data from cohort 1 do not contribute significantly to the message in this manuscript.

Response: Although the NEJM study shows this for immune competent persons, we respectfully feel that it is important to assess this in transplant recipients who are significantly immunocompromised. Interestingly, although heterotypic responses to BA.1 were low, there were some patients who demonstrated cross-reactive antibody neutralization and polyfunctional BA.1 specific T-cell responses. What is new and interesting though, is that based on our additional analysis of BA.2 samples in the entire cohorts, we have observed that in cohort 1, heterotypic neutralization against BA.2 seems to be somewhat better than heterotypic neutralization against BA.1. We have added this to the results section.

3. There are missed opportunities to explore correlations between T cell and antibody responses. Were there subjects that made poor antibody responses, but had good T cell responses? Are these two aspects of the immune response linked as tightly in this immunocompromised population as previously thought? What about the outliers that failed to make T cell responses? Are there specific demographic or clinical factors that explain this lack of response?

Responses: We agree. Now that we have performed additional T-cell testing, we have gone ahead and done the above requested analyses. Correlative analysis of T-cell response and BA.1 neutralizing antibody response for cohort 2 (omicron infected) is now added. In addition, we have looked at whether any demographic factors predicted the T-cell response in Cohort 2. A table comparing clinical factors between those with and without a BA.1-directed T-cell responses are show in supplementary Tables 2 and 3. We specifically chose polyfunctional CD4+ T-cells and CD8+ T-cells as these cells are commonly detected in immunogenicity studies and are thought to be functionally superior to monofunctional T-cell responses. It also minimizes the number of statistical comparisons to avoid over-interpretation of data.

Minor concerns:

-time from events are very heterogenous, this should be addressed or controlled for in some way

Response: We believe the reviewer is referring to the timing of infection/vaccination as it relates to the date of transplant. We acknowledge that there was some heterogeneity with some patients earlier post-transplant and others later. However, as recommended below as well, we have analyzed this variable in the three transplant cohorts, and no statistically significant difference was observed (Supplementary Table 1). However, we have also now acknowledged this heterogeneity as a limitation in the discussion.

-what is the source of the 293 cells and the 293-ACE2/TMPRSS2 cells?

Response: HEK293T cells (not HEK293 cells) were purchased originally from ATCC (ATCC #CRL-3216) and HEK293T-ACE2/TMPRSS2 cells were prepared in the lab according to a previously published protocol. This has been added to the methods.

-lack of consistency regarding the naming of the variants (strains vs. variants), not sure which is correct, but should be consistent

Response: We have reviewed the manuscript and have made efforts to use more consistent terminology throughout the paper (removed the word strain; used more precise terminology) .

-additional current literature on this topic should be cited and the current study put in context of these other studies (not all need to be included, but some suggestions):

10.1016/j.jhep.2022.03.042, 10.1097/TP.0000000000004140,
10.1001/jamanetworkopen.2022.6822, <https://doi.org/10.1093/infdis/jiac118>,
10.1126/scitranslmed.abl6141

Response: We thank the reviewer for bringing these papers to our attention. We have added discussion on several of these papers to the revision, where appropriate. Please see lines 371-381

-the term “wild-type” virus is imprecise and implies these variants were somehow modified rationally. Is this D614G? Wuhan-1? WA-1? Vaccine spike? Would replace the term wild-type with something more specific

Response: To provide more precise wording, we have replaced references to “wild-type” with ancestral SARS-CoV-2 or D614G, where appropriate.

-there is no such thing as a “COVID-19 infection.” It is a SARS-CoV-2 infection.

Response: We agree and have ensured we do not have this term in the paper.

-data on how the authors determined what variant infected each patient is not clear. Were the viral genomes sequenced? There are no methods on this.

Response: Variant determination of samples was performed using C19-SPAR-Seq (Systematic Parallel Analysis of RNA coupled to Sequencing), a multiplexed amplicon-based, scalable, automated sequencing platform for SARS-CoV-2 variant detection and identification. In comparison to whole genome sequencing (WGS), C19-SPAR-Seq has a >95% sensitivity. Additional details can be found in the following reference: Aynaud, Hernandez, Barutcu et al., 2021; Nat Commun. PMID: 33658502. Testing was done in the clinical lab which regularly validates results against whole genome sequencing. More details have been added to the methods.

-typo in the sentence “Of the recruited patients, typing was available in 20 patients and and confirmed Omicron BA.1 infection in all cases.”

Response: Thank you for bringing this duplication to our attention. We have corrected this; please see line 183.

-where are the stats comparing the demographic and clinical aspects of the various cohorts in Table 1? To that end, it appears cohort 3 is on average closer to transplant? Could that partially explain the poorer response?

Response: We have added a Supplementary table 1 with the association measures between baseline characteristics of each of the 3 transplant cohorts. There is a significant difference in the age of the cohorts; however the type of transplant and the immunosuppressive regimen are similar among the three cohorts. The time from transplant is not significantly different between the three cohorts. Based on this we believe the cohorts to be comparable. A multivariate regression analysis including age, immunosuppression, and time from transplant did not change the significance of the findings related to neutralizing antibody (data not shown).

-In Figure 2a where is the LLOD indicated?, shouldn't this be Log3 transformed given the authors performed serial 3-fold dilutions (Figure 3A too)

Response: The log transformation is somewhat arbitrary but is typically log10 for ease of interpretation and consistency with the reported literature. The LLOD for the ID50 is defined as those patient sera with a calculated absence of 50% neutralization with undiluted serum; this equates to a \log_{10} ID50 of zero and was considered the threshold for lower limit of detection. This is outlined in the statistical methods but we have now added further explanation in the methods section (statistical analysis) as well for clarity.

-figure 2b is not compelling. Why only test 10 samples? It seems like there are a couple of outliers that are really skewing things

Response: As indicated above, additional testing on all samples has now been performed to eliminate bias in subset selection and testing. Figure 2b has been revised.

-how were the 25 subjects in figure 1c and figure 1d chosen? How representative of cohort 1 are they (see major concerns above)?

Response: For cohort 1, PBMCs were collected only in subsets of patients. Earlier in the pandemic, there were challenges to specimen collection from COVID positive patients. Also some patients chose only to provide blood for serology testing. The 25 subjects appear to be representative of the cohort as we found no significant difference in terms of age, sex, type of transplant and immunosuppressive regimen between the 25 patients and the remaining cohort (data not shown). For cohort 2 (omicron infected) we were able to collect PBMC samples

more consistently and so now as outlined above we have done T-cell testing on the majority of patients in cohort 2 to address the previously raised concerns.

-is there anything special about the people that DID have CD8 positive T cells? Non-kidney? Immunocompromised regimen (see major concerns above)?

Response: See previous response. We have now looked more carefully at correlates of CD4 and CD8 Tcell responses in the omicron infected cohort. See supplementary Tables 2 and 3

-the anti-N data demonstrating that patients weren't infected should be shown.

Response: We have now included the anti-NP data in a supplementary figure S2.

-in figure 4a a Kruskal-wallis test before testing individual group comparisons would be appropriate

Response: As suggested, we performed a Kruskal-Wallis test prior to individual group comparisons in Figures 4A and 4C. P-values were statistically significant indicating the medians varied between groups for both neutralizing antibodies and T-cells ($p < 0.001$ for both). These details have also been added to the results section.

-in figure 4c, are these all the data?
-where are the CD8s in Figure 4?

Response: We have now added CD8 T-cells to Figure 4 specifically focusing on polyfunctional cells.

-In Figure 5B when were these data collected relative to infection (and possible mAb treatment)?
Are you actually measuring sotrovimab in these plasma samples?

Response: This is a good point (also noted by reviewer 1). As above, in order to try and assess this possibility, we looked at the impact of Sotrovimab vs. no-sotrovimab on the detection of neutralizing antibody in this cohort. We did not see any significant difference (Figure 5B) suggesting that Sotrovimab itself did not have a major influence on the subsequent level of neutralization detection. The median time from COVID-19 diagnosis to sample collection was 40 days (IQR 36-46) and patients typically received Sotrovimab 1-2 days following diagnosis. Therefore significant time did elapse between Sotrovimab and sample collection; it may be possible that in the presence of illness the half-life of sotrovimab may be too short to allow detection this long after administration. This point has been added to the results.

-Figure 6a needs Kruskal-Wallis test

Response: A Kruskal-Wallis test was performed and found to be statistically significant, $p < 0.001$. This data has been added to the paper including in the results section. Please see Line 335.

-Figure 6 B – E should have the same X-axis scale

Response: We have adjusted Figure 6B-E to ensure it has the same axes as the others.

-conclusion about vaccine + infection (non-omicron) being insufficient is not supported by the data because cohort 1 is largely unvaccinated

Response: We have adjusted the wording in the conclusion to more clearly reflect this point.

REVIEWERS' COMMENTS

Reviewer #1 (Remarks to the Author):

The authors present a revision of their manuscript wherein they have added further testing for BA.2 neutralization which provides further insight into a common circulating strain, and offer some suggestion of cross-protection between BA.1 and BA.2 immune responses. This also provides a helpful baseline for future studies on BA.4 and BA.5 which the authors may consider. They have addressed many of the revisions and limitations outlined in the prior review appropriately. Below are several additional considerations.

Line 316 - the projected half life of sotrovimab is 49 days, there is consideration from the company for using it for pre-exposure prophylaxis due to the long half-life. I suspect there would still be significant levels of antibody even 40 days out which could otherwise affect the neutralization responses. The analysis of responses being similar between those who received and did not receive sotrovimab is helpful, but could be confounded by bias in selecting patients at higher risk for complications being the ones to receive sotrovimab. This may be a limitation that cannot be avoided due to the need to treat many of these breakthrough cases, but may nevertheless influence the results. However, the T-cell responses in this cohort remain suggestive that this is not entirely passive immunity. This same issue may be noted with receipt of tixagevimab-cilgavimab in future studies.

Table 1 - It may be worth mentioning specifically that no patients received B-cell depletion either in the table or text.

Figure 1 text - should n=71 for ancestral strain? Would also change phrasing to "infected during Omicron wave" as you have in other places in the text.

Reviewer #2 (Remarks to the Author):

I thank the authors for their careful and thoughtful improvements to their manuscript. The inclusion of additional BA.2 data is particularly compelling. The additional clarity provided on the cohorts also allows for a more nuanced and accurate assessment of their findings. I have only two remaining concerns.

1. In this era of nearly limitless supplemental data files, I do not think it is appropriate to say "data not shown." So I believe the comparison of the 25 patients in cohort 1 to the rest of cohort 1 (line 221) should shown in the supplements.

2. Kindly provide a reference for the statement made regarding the superiority of polyfunctional T cells (line 265).

REVIEWERS' COMMENTS

Reviewer #1 (Remarks to the Author):

The authors present a revision of their manuscript wherein they have added further testing for BA.2 neutralization which provides further insight into a common circulating strain, and offer some suggestion of cross-protection between BA.1 and BA.2 immune responses. This also provides a helpful baseline for future studies on BA.4 and BA.5 which the authors may consider. They have addressed many of the revisions and limitations outlined in the prior review appropriately. Below are several additional considerations.

Response: We thank the reviewer for these positive comments

Line 316 - the projected half life of sotrovimab is 49 days, there is consideration from the company for using it for pre-exposure prophylaxis due to the long half-life. I suspect there would still be significant levels of antibody even 40 days out which could otherwise affect the neutralization responses. The analysis of responses being similar between those who received and did not receive sotrovimab is helpful, but could be confounded by bias in selecting patients at higher risk for complications being the ones to receive sotrovimab. This may be a limitation that cannot be avoided due to the need to treat many of these breakthrough cases, but may nevertheless influence the results. However, the T-cell responses in this cohort remain suggestive that this is not entirely passive immunity. This same issue may be noted with receipt of tixagevimab-cilgavimab in future studies.

Response: This is a valid concern. However, in our clinical practice all transplant patients (since all are on exogenous immunosuppression) are deemed high risk for progression/complications and as such have access to sotrovimab therapy as long as they presented within 7 days of symptoms and were able to travel to the infusion clinic. Therefore, this type of selection bias generally did not play a role in treatment decisions.

Table 1 - It may be worth mentioning specifically that no patients received B-cell depletion either in the table or text,.

Response: This statement had been added to the Results. Please see line 252-253 of revision document.

Figure 1 text - should n=71 for ancestral strain? Would also change phrasing to "infected during Omicron wave" as you have in other places in the text.

Response: Yes, thank you for identifying this error. It should be n=71, not n=17 for ancestral SARS-CoV-2 in the figure 1 legend.

We have rephrased the figure 1 legend as suggested.

Reviewer #2 (Remarks to the Author):

I thank the authors for their careful and thoughtful improvements to their manuscript. The inclusion of additional BA.2 data is particularly compelling. The additional clarity provided on the cohorts also allows for a more nuanced and accurate assessment of their findings. I have only two remaining concerns.

Response: We thank the reviewer for these positive comments

1. In this era of nearly limitless supplemental data files, I do not think it is appropriate to say "data not shown." So I believe the comparison of the 25 patients in cohort 1 to the rest of cohort 1 (line 221) should shown in the supplements.

Response: As suggested we have added a supplemental table with the comparison requested above (Supplementary Table 2).

2. Kindly provide a reference for the statement made regarding the superiority of polyfunctional T cells (line 265).

Response: We had added references for two reviews where polyfunctional T-cells are defined and discussed in the context of whether they are superior to monofunctional T-cell responses.